# Challenges of Retrieving LULC Information in Rural-Forest Mosaic Landscapes Using Random Forest Technique

Chinsu Lin [1,*] and Nova D. Doyog [1,2]

1   Department of Forestry and Natural Resources, National Chiayi University, 300 University Road, Chiayi 600035, Taiwan
2   College of Forestry, Benguet State University, La Trinidad 2601, Benguet, Philippines
*   Correspondence: chinsu@mail.ncyu.edu.tw

**Abstract:** Land use and land cover (LULC) information plays a crucial role in determining the trend of the global carbon cycle in various fields, such as urban land planning, agriculture, rural management, and sustainable development, and serves as an up-to-date indicator of forest changes. Accurate and reliable LULC information is needed to address the detailed changes in conservation-based and development-based classes. This study integrates Sentinel-2 multispectral surface reflectance and vegetation indices, and lidar-based canopy height and slope to generate a random forest model for 3-level LULC classification. The challenges for LULC classification by RF approach are discussed by comparing it with the SVM model. To summarize, the RF model achieved an overall accuracy (OA) of 0.79 and a macro F1-score of 0.72 for the Level-III classification. In contrast, the SVM model outperformed the RF model by 0.04 and 0.09 in OA and macro F1-score, respectively. The accuracy difference increased to 0.89 vs. 0.96 for OA and 0.79 vs. 0.91 for macro F1-score for the Level-I classification. The mapping reliability of the RF model for different classes with nearly identical features was challenging with regard to precision and recall measures which are both inconsistent in the RF model. Therefore, further research is needed to close the knowledge gap associated with reliable and high thematic LULC mapping using the RF classifier.

**Keywords:** forest classification; mapping; forest degradation; agriculture; machine learning; sustainability

## 1. Introduction

Capturing and reporting the status of earth's land use and land cover (LULC) is indispensable in reducing emissions from deforestation and degradation (REDD) and in mitigating global warming as it plays a crucial role in determining the trends of the global carbon cycle [1–4]. Areas of forest coverage, healthy and structural components such as forest types and species, are critical information for forest monitoring and sustainable management [5]. LULC conversion is a complex process that involves anthropogenic activities and biological, environmental, and meteorological factors, and could also directly or indirectly affect its environment in terms of climate, global climate change patterns, services such as economy, biodiversity, forest growth, food, and water cycle [6,7], and aesthetic and economic value [8]. Change in LULC could hasten the risk of natural hazards due to the loss of protective services that LULC offers and could also lead to increased fragmentation and structural destruction of protected areas [9]. Diverse degrees of LULC changes can significantly affect the ecosystem as this interferes with natural ecological processes, such as nutrient cycling, water cycle, energy flow, and succession. The changes that provide critical information to address and control the impacts should be measured and assessed. From the point of view of forest resources assessment, forest inventory must be reliable in measurement, report, and validation [10]. Derivation of accurate LULC to disclose details of forest status is of particular importance in achieving sustainable management.

Accurate mapping of LULC is highly related to the resolutions and features inherited from the diverse remotely sensed data delivered by spaceborne, airborne, and drone

platforms. In terms of spectral features, multispectral satellite data with high spatial resolution at a low cost of data acquisition appears to be a major application. Although much research demonstrated the advantages of global and regional mapping, the hundred-meter scale resolution of LULC products has the problem of information uncertainty. Fortunately, image fusion techniques extend the capability of retrieving high spectral resolution and high spatial resolution from the satellite data, providing the opportunity to produce accurate LULC maps to support constant monitoring of changes in forest ecosystems [11]. In contrast to the MODIS with significant low spatial resolution and SPOT data with a large view angle of observation, Landsat and Sentinel-2 sensors collect a broad range of visible-NIR-SWIR spectra. In a nadir-view, operational mechanisms become more convenient and preferable with a view to LULC mapping. This is more evident when shadow covers significant areas of the whole image because minimizing impacts of shadow-induced abnormal variations of spectral features to derive surface materials' properties is still a considerable challenge. LULC classification accuracy is most likely questionable due to the increasing shadow caused by the sensor's view angle and terrain morphology interactions.

As disclosed in various articles, the problems being encountered in LULC classification or forest species inventory include low variability of the spectral profile of the different LULC, and high intra-class variability [12,13]; noises embedded in satellite data, low spatial resolution of freely acquired satellite data especially when working at rural level, and lack of classifiers that can be easily interpreted and automated. Recently, machine-learning techniques, including decision trees, neural networks, support vector machines, and random forest, have been widely applied to Landsat [14,15], SPOT [16], MODIS [17–19], and Sentinel-2 imagery [20–23]. Recently, deep learning techniques have been intensively explored in diverse applications, such as pest detection [24], segmentation [25,26], species classification [27], and LULC classification [28]. Although deep learning techniques are excellent in retrieving multidimensional features to recognize, detect, and segment objects for labeling, it is challenging to describe how deep learning techniques deal with multiple features to achieve the work. Therefore, machine learning techniques such as random forest (RF) and support vector machine (SVM) are more appropriate. From the existing and numerous classifiers, the performance of random forest has already proved superior in comparison to other methods [29–31] regardless of the source satellite data such as multispectral and hyperspectral satellite sensor imagery [13,32–41]. Random forest requires fewer parameters and minimal manual intervention in high-dimensional data processing; it can rapidly obtain classification results with high accuracy [32,37,38].

In the practical management of forest ecosystems, developing an appropriate strategy to minimize competition for land or resources between agriculture and forest services [42,43] to support anthropogenic needs and sustainable forestry has always been a key issue in recent decades. The Ex-Mega Rice Project (EMPR) of Indonesia, 1997–1999, is a typical example showing inappropriate land use planning damage to the forest ecosystem; the project attempted to increase agricultural production but ultimately led to a devastating loss of forest resources and biodiversity in the Kalimantan area [44]. As noted, detailed spatial-explicit information of species and forest-type composition over forest lands helps diagnose areas of forest degradation and deforestation. Classifying LULC with remotely sensed data faces the challenge of differentiating conservational and degrading-oriented classes. Specifically, the similarity of spectral features in the areas of well-covered tree species/forest type, crops, and grasses, as well as the areas with homogeneous features of bare soil in flat land and eroded land. A classification that concerns only primary materials, such as vegetation, built-up, soil, and water, is insufficient to provide detailed information for planning with regard to a rural landscape. In other words, forest inventory-oriented LULC mapping is indeed required to retrieve detailed attributes of LULC from satellite images by appropriate classifiers to support diagnoses of agricultural expansion, forest degradation, and disturbance [45,46], temporary updates of land cover maps [47], and the

drivers of forest changes [48,49]. This kind of management-oriented LULC classification issue has been rarely explored in the literature.

As noted, many studies also suggested that appropriate classifiers in combination with good-quality data are crucial in LULC classification. Therefore, this study aims to determine the supplementary abilities of the well-performing RF technique in establishing forest composition-oriented LULC information. Information classes are the end-members frequently observed in agriculture-forest mosaic landscapes whose occurrence and distribution can disclose the condition related to forest management, forest degradation, and disturbances. Confusion in the end-members classification is considered uncertainty. The SVM technique was also applied to explore the uncertainty and challenges in the 3-level LULC classification.

## 2. Materials and Methods

### 2.1. Study Site

Lishan, located in Heping, Taichung (Figure 1), is part of the Jade Mountains of Taiwan. The study site, which was delineated within the boundary of the site for the fuel project conducted by [50], has a total area of 20,288 ha and has an average altitude of 2001 m above sea level (masl) with a minimum of 1386 and a maximum of 3088 masl. The standard deviation of the elevation in the study site is 326.79 m. The temperature of the area ranges from 24° during summer to −4° Celsius during winter. Lishan is a hub connecting the east, west, and north and south in Central Taiwan. It is surrounded by Taroko National Park to the east, Shei-Pa National Park to the west, Wushe to the south, and Taipingshan National Forest Recreation Area to the north. Due to its accessibility and scenic sites, Lishan became a getaway for holidays and weekends to escape the island's summer heat. However, it is also threatened by forest disturbances, such as fire and agricultural expansion [50]. For instance, an average of 526 ha per year were destroyed by fire within and around the area from 1963 to 2019 [50]. Lishan is noted for producing pears, apples, peaches, and tea, and for its mountainous scenic view.

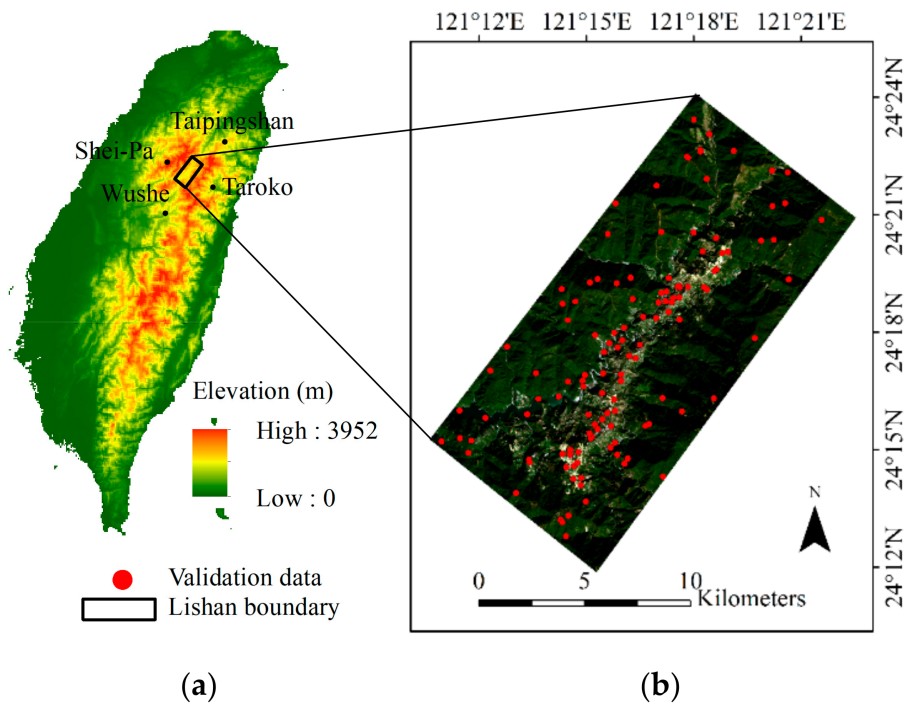

(**a**)                                                                          (**b**)

**Figure 1.** The digital elevation model (DEM) of Taiwan overlaid with the boundary of the study site (**a**); and geolocations of the validation dataset overlaid with a Sentinel-2 natural-color image (**b**).

### 2.2. Data Processing and LULC Classification

Two Sentinel-2B L1C top-of-atmosphere digital number images with the ID of L1C_T51QUG_A014086_20191117T022951 and L1C_T51RUH_A014086_20191117T022951 acquired on 17 November 2019, were downloaded freely from the open-access hub of the ESA-Copernicus (https://scihub.copernicus.eu/ (accessed on 16 February 2021)). As shown in Figure 2, the images were first atmospherically corrected to retrieve bottom-of-atmosphere surface reflectance through the sen2cor utility [51] and then pansharpened using the Sen2Res algorithm [52] to generate a spatially registered multispectral dataset of 10-m resolution. The bands with a lower resolution of 20 m or 60 m were super-resolved based on the geometry and reflectance consistency of neighboring pixels between the higher and lower resolution bands. Next, every band of the super-resolved image was layer-stacked. Finally, the two preprocessed Sentinel-2 images were mosaicked, subset, and subjected to LULC classification.

Information on LULC from remotely sensed images is mainly interpreted via spectral characteristics. Excluding built-up areas, vegetation, bare soil, and water are major land cover classes in nature. Since both the chemical and physical properties of the components tend to make the pattern of the reflectance curve highly variable, mapping accurate and reliable LULC classes remains challenging [53]. A combination of appropriate image normalization and classification techniques can likely differentiate the nature of land covers. It is, therefore, the key to deriving precise LULC information at the landscape and national levels. In remote sensing, a difference in reflectance indicates the dissimilarity of the pixels; in addition, a change of reflectance in multi-temporal images for the exact location reveals the conversion of the components. Since the properties of vegetation and bare soil control the depth of classes among the levels of classification, this study utilized the bare soil index (BSI) [54], modified difference normalized water index (MNDWI) [55], and normalized difference vegetation indices (NDVI) to help in the differentiation of reflectance profile of the LULC classes. The vegetation indices were generated from the SR image and collectively coded as Sentinel-2 optical metrics.

The integration of the height information during the LULC classification allows the separation of grasses or short vegetation from bare soil, medium and high vegetation, and buildings. Slope information provides the steepness of a particular area. Object height and terrain slope information were derived from airborne lidar scanner (ALS) point cloud data which were acquired in December 2018 using the Riegl LMS-Q780 lidar system [50]. During the data collection, the operating flight altitude ranged from 3400–4000 m and the laser pulse repetition rate was 240–270 kHz. Briefly, the original point cloud data which has a ground and canopy point cloud density of ~2.5 and ~15 points/m$^2$, respectively, were first subjected to denoising and classification, then sent to generate a digital elevation model (DEM) and digital surface model (DSM) with a resolution of 1-m size using linear interpolation technique. Next, the canopy height model (CHM) was derived by subtracting the DEM from the spike-free DSM. The slope was generated from the DEM. The CHM was then subjected to pitfall removal using the filtering method for object height determination. Alternatively, CHM data can also be derived from spaceborne full-waveform lidar data such as geoscience laser altimeter system (GLAS) and global ecosystem dynamics investigation (GEDI), and advanced topographic laser altimeter system (ATLAS)—a photon-counting lidar [56,57]—when integrating with moderate-resolution images. Though GLAS, GEDI, and ATLAS lidar can only provide footprint information, wall-to-wall CHM can be derived through regression techniques and with ancillary data. As a result, a layer-stacked image with 17 features, including 12 spectral features (the cirrus band was excluded), three vegetation indices, and two lidar-derived metrics, was generated for this study. The continuous values of all the data were used in the LULC classification. The Sentinel-2 SR range from 0 to 0.41 across the bands and the values of BSI, MNDWI, and NDVI derived from the Sentinel-2 SR image were between −1 and +1. Meanwhile, the values of the ALS-derived slope and CHM of the study ranged from 0%–4806% and 0–63.74 m, respectively. As supplementary data, the orthoimage of the study site, which was collected

simultaneously with the ALS point clouds, was used purposely for visualization and verification purposes of the ground truth data. The overall flowchart of the data collection and processing until the LULC classification and validation is summarized in Figure 2.

Based on the USGS LULC system for use with remote sensor data [58], the six types of Level-I LULC represented in this area are forest, grassland, agriculture, built-up, bare land, and water, which are categorized further into 10 Level-II LULC and a total of 13 Level-III LULC types namely building, broadleaf, pine, cypress, other conifers, orchard, tea farm, vegetated cropland and non-vegetated cropland, grassland, water, sand, and eroded land. Table 1 shows the hierarchical classification of the LULC classes in this study. Figure 3 demonstrates the overall conceptual model of the random forest classification. To implement the LULC classification by random forest method [59], the parameters used were 16 features and 500 decision trees as the lowest error was reached at this combination. A sum of 26,769 sample pixels covering the 13 Level-III LULCs were randomly collected. Through simultaneous visual inspection of the orthoimage and Sentinel-2 image of the study site, representative samples were collected. In cases of conflicting classes of the same pixel, e.g., vegetated cropland in the orthoimage but non-vegetated cropland in the Sentinel-2 image, the prevailing class in the Sentinel-2 image was followed. The collected samples were then randomly separated into two datasets: 80% for modeling and the remaining 20% for validation. The 13 Level-III LULCs were then aggregated into 10 Level-II LULCs, and finally into 6 Level-I LULCs based on the class's attributes. The aggregation of classes was based on the hierarchical level as is presented in Table 1. To compare the performance of the RF models with other high-performing machine learning algorithms, the LULC classification was again performed but this time, using the SVM classifier. The SVM modeling was implemented via the radial basis function. Using the 20% validation data, the classification accuracy of the LULC products was evaluated using the recall, precision, and F1-score from the viewpoint of particular classes and overall accuracy (OA), kappa coefficient (κ), and the macro F1-score for the general measures [28]. The class-based accuracy indices are further described in Equations (1)–(3). The OA is the ratio of the total number of correctly predicted classes to the total number of validation data while the macro F1-score is calculated as the average of all the per-class F1-scores.

$$\text{Recall} = \frac{\text{True positives}}{(\text{True positives} + \text{False negatives})} \tag{1}$$

$$\text{Precision} = \frac{\text{True positives}}{(\text{True positives} + \text{False positives})} \tag{2}$$

$$\text{F1-score} = 2 \times \left[ \frac{(\text{Recall} \times \text{Precision})}{(\text{Recall} + \text{Precision})} \right] \tag{3}$$

In Equations (1)–(3), true positives are the observations that were correctly predicted by the model, false positives are those predicted data that were included in a class but do not actually belong to the class, and false negatives are those that were removed from a particular class and included wrongly in a different class. The class-based error matrix was presented in probability values instead of the traditional method which uses the actual number of samples.

**Table 1.** Hierarchical classification of the Level-III, -II, and -I LULC classes.

| Level-III | Level-II | Level-I |
|---|---|---|
| 1. Broadleaf (Br) | 1. Br | |
| 2. Cypress (Cy) | | |
| 3. Pine (P) | 2. Conifer (Co) | 1. Forest (F) |
| 4. Other Conifers (OC) | | |
| 5. Grassland (Gl) | 3. Gl | 2. Gl |

**Table 1.** *Cont.*

| Level-III | Level-II | Level-I |
|---|---|---|
| 6. Orchard (Or) | 4. Or | |
| 7. Tea farm (TF) | 5. TF | |
| 8. Vegetated cropland (Clv) | | 3. Agriculture (Ag) |
| 9. Non-vegetated cropland (Clb) | 6. Cropland (Cl) | |
| 10. Built-up (Bu) | 7. Bu | 4. Bu |
| 11. Eroded land (EL) | 8. EL | |
| 12. Sand (Sa) | 9. Sa | 5. Bareland (Bl) |
| 13. Water (W) | 10. W | 6. W |

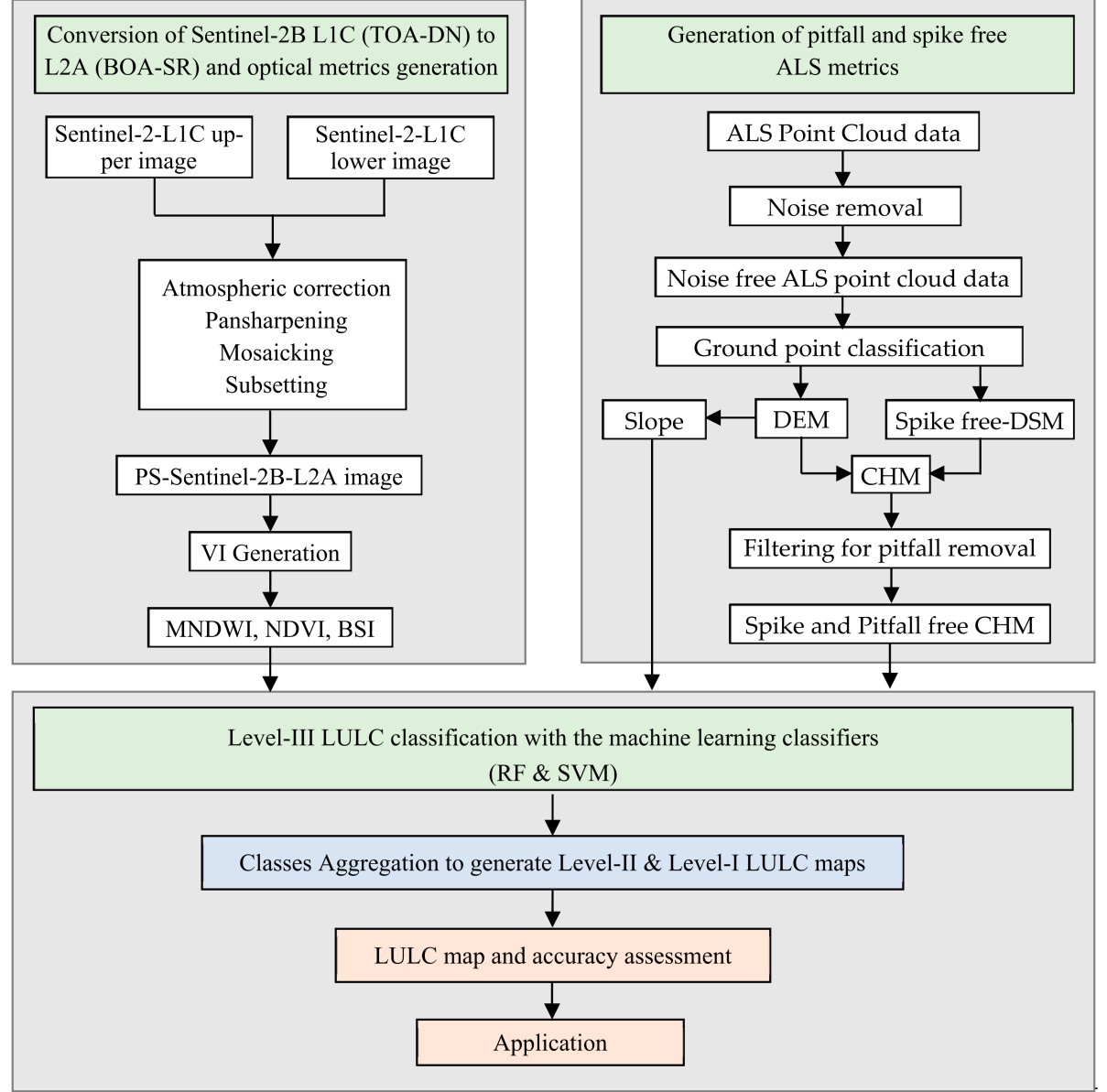

**Figure 2.** Flowchart of integrating Sentinel-2 and ALS data for LULC classification via RF and SVM machine learning technique. The classifications were performed using the package randomForest of [60] in R statistical software (version 4.1.2) and the SVM tool in ENVI software (version 5.5.3).

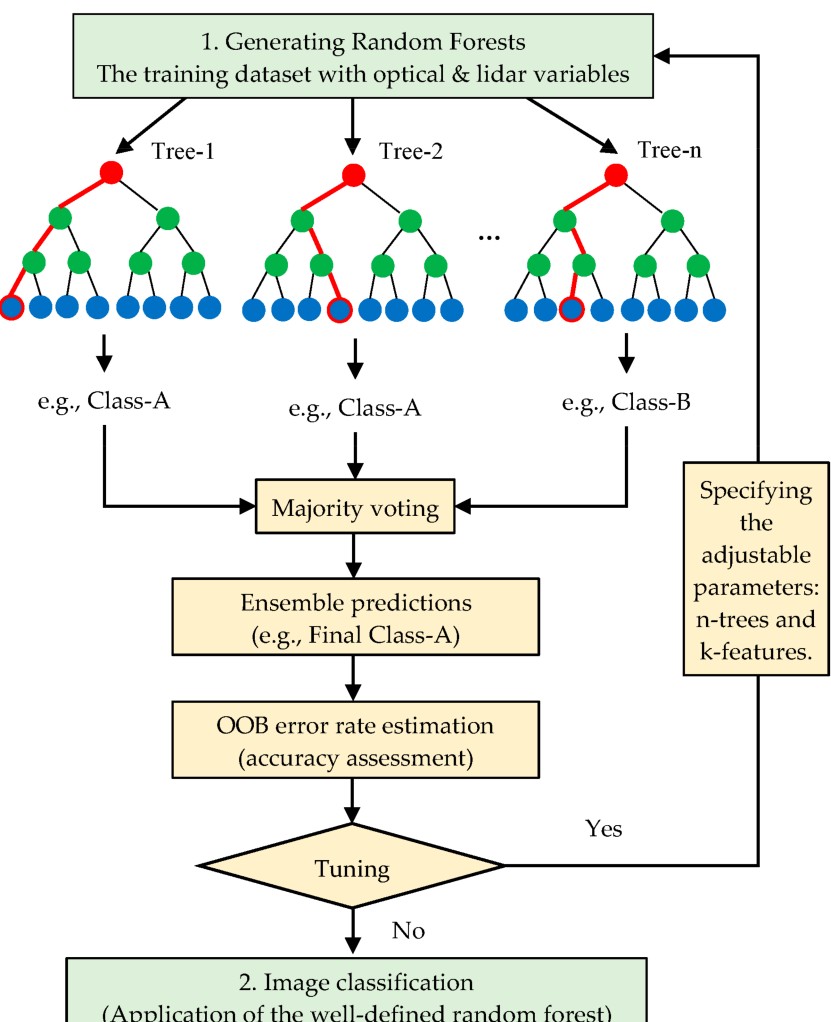

**The RF conceptual model (parameters and criteria)**

1. Generating uncorrelated decision trees by the feature-randomness rule
   - Bagging with replacement
     - Bagged (training) vs. OOB (testing)
     - Randomly selected n-trees with k-features
   - Splitting every decision tree
     - Splits the node on all features and selects the split with the lowest value of Gini impurity
     - Repeat splits until Gini impurity = 0, which indicates that homogeneous child nodes are achieved
     - Structure of a decision tree
       🔴Root 🟢Decision 🔵Leaf nodes
   - Prediction by every single decision tree
   - Majority voting to ensemble predictions (determining the outcome of predictions)
   - OOB error estimation
   - RF tuning (if OOB error rate > 0)
     - Removing the least important feature based on the criteria MDA & MDG
     - An RF with the lowest OOB error rate is determined.
2. Forming the well-defined random forest for the final application
   - Predictions to label a LULC for the pixels of an image with all available features

**Figure 3.** The processes of the random forest classifier in generating independent decision trees for LULC mapping. The highlighted red path from the root node through the decision node to the leaf node demonstrates the prediction of that decision tree. The final result is the ensemble predictions based on the majority of the prediction made by all the trees in the random forest.

## 3. Results

### 3.1. Generalized Spectral Features of LULCs

In remote sensing, surface reflectance is usually used to describe the spectra behavior of end-members. As shown in Figure 4, the reflectance curves are generalized from training samples of the Sentinel-2 SR images. The curves can be characterized or grouped into three particular patterns: double peaks in visible and infrared regions such as vegetation classes, a single peak in the infrared region such as bare-soil-based classes, and a single peak in the visible region such as water. Obviously, these three primary categories appeared distinguishable through the spectral features from the visible to near-infrared and shortwave infrared. Apparently, the surface reflectance of tea farm and vegetated cropland differs from those forest-relevant classes. Additionally, the curve trend in the green-red edge wavelengths highlighted in Figure 4b also shows a positive opportunity to differentiate grassland, vegetated cropland, and orchard from each other and even forest-relevant classes. Unfortunately, the separability in generalized spectral curves of vegetation LULC classes became ambiguous or unclear when considering the variance of spectral reflectance. This is evident in Figure 5, which demonstrates spatial and spectral variations of the Level-III LULC classes in a tabular form with very high-resolution orthoimages, high-resolution Sentinel-2 SR images, and correspondingly their VNIR-SWIR spectral reflectance curves.

The orthoimage and Sentinel-2 image provide spatial features of the classes at a scale of 0.2 m and 10 m, and the reflectance curve shows the generalized spectral features of the LULC classes. The spectral pattern of vegetative and non-vegetative classes is distinct, allowing us to differentiate them through Sentinel-2 SR image.

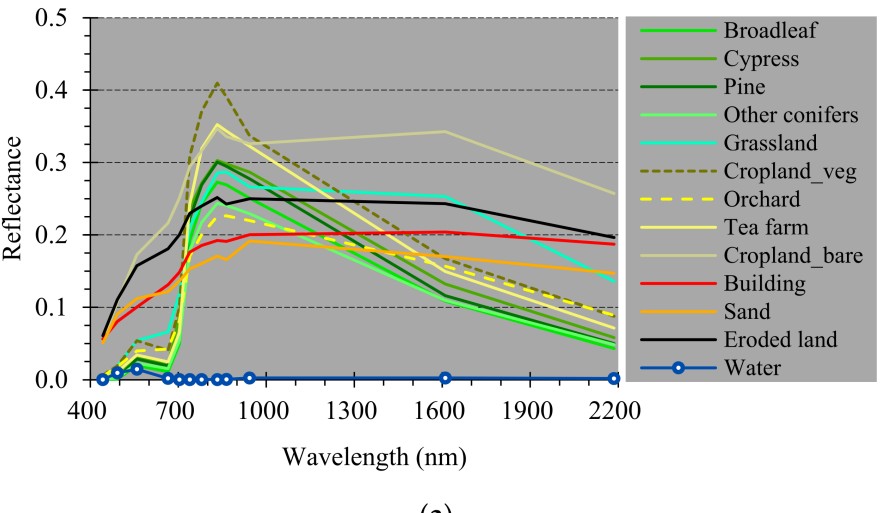
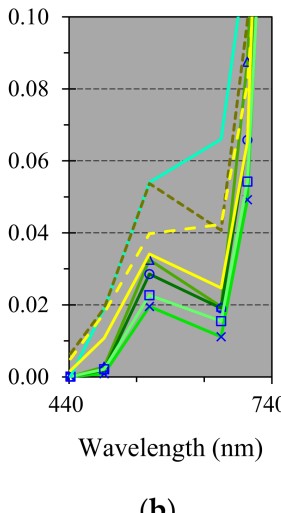

(**a**) (**b**)

**Figure 4.** Spectral features of the end-members for Level-III LULC classes: (**a**) the line graph combines the spectral curve of each class for better comparison in which, the circles that come across the water reflectance curve indicate the central-wavelength locations of the Sentinel-2 multispectral bands (No. 1-8, 8a, 9, 11, 12) except the cirrus band (No. 10 center at the wavelength 1375 nm) that is excluded in the classification; (**b**) the enlarged line graph of visible wavelengths in subfigure (**a**) highlights the surface reflectance differences among the eight vegetation classes.

In contrast to the non-vegetative LULCs, the vegetative classes show a similar reflectance curve over the spectral region. Although the average reflectance of those classes appeared with different levels in Figure 4, reflectance between classes is overlapped when considering their significant standard deviation as shown in rows three and six of Figure 5. Looking at the enlarged map to the right of Figure 4, the spectral patterns of the grassland, vegetated cropland, and orchard are quite different from the others. The reflectance of vegetated cropland in wavelengths 440–559 nm acts almost identically to the grassland while the former drops lower, but the latter raises significantly at the wavelength 665 nm. This kind of different reflectance behavior increases the opportunity to differentiate classes.

Based on the region-of-interest areas for generating RF models, the average metric value of each class is presented in the line graph in Figure 6. For visualization purposes, the CHM and slope are rescaled to 0–1 by dividing a rescale factor (SF) of 22.41 and 107.35, respectively. The SF is the maximum average value of the CHM and slope of the different LULC classes. As can be seen in the CHM map in Figure 6, lower rescaled values of CHM are mainly distributed along with the river system and agricultural areas next to the river while higher values are distributed over the forest area. This is evident in the line graph where the forest-relevant classes are far from zero. In contrast, the agriculture-relevant classes are quite close to zero. In addition, the ALS-derived feature slope is of particular importance to differentiate the eroded land from the non-vegetated cropland, sand, and building. Furthermore, the NDVI of those non-vegetation classes is so small that can be recognized and separated from vegetation classes. All the classes have negative MNDWI values except for the water which has a positive value. The vegetation has lower MNDWI values than the non-vegetation. Building and open land classes, the eroded land, sand, and bare cropland, have the highest BSI, and the others are with negative BSI values. To summarize, the dissimilarity of spectral and lidar metrics among LULC

classes offers additional opportunities for generating decision trees of the RF model for better classification.

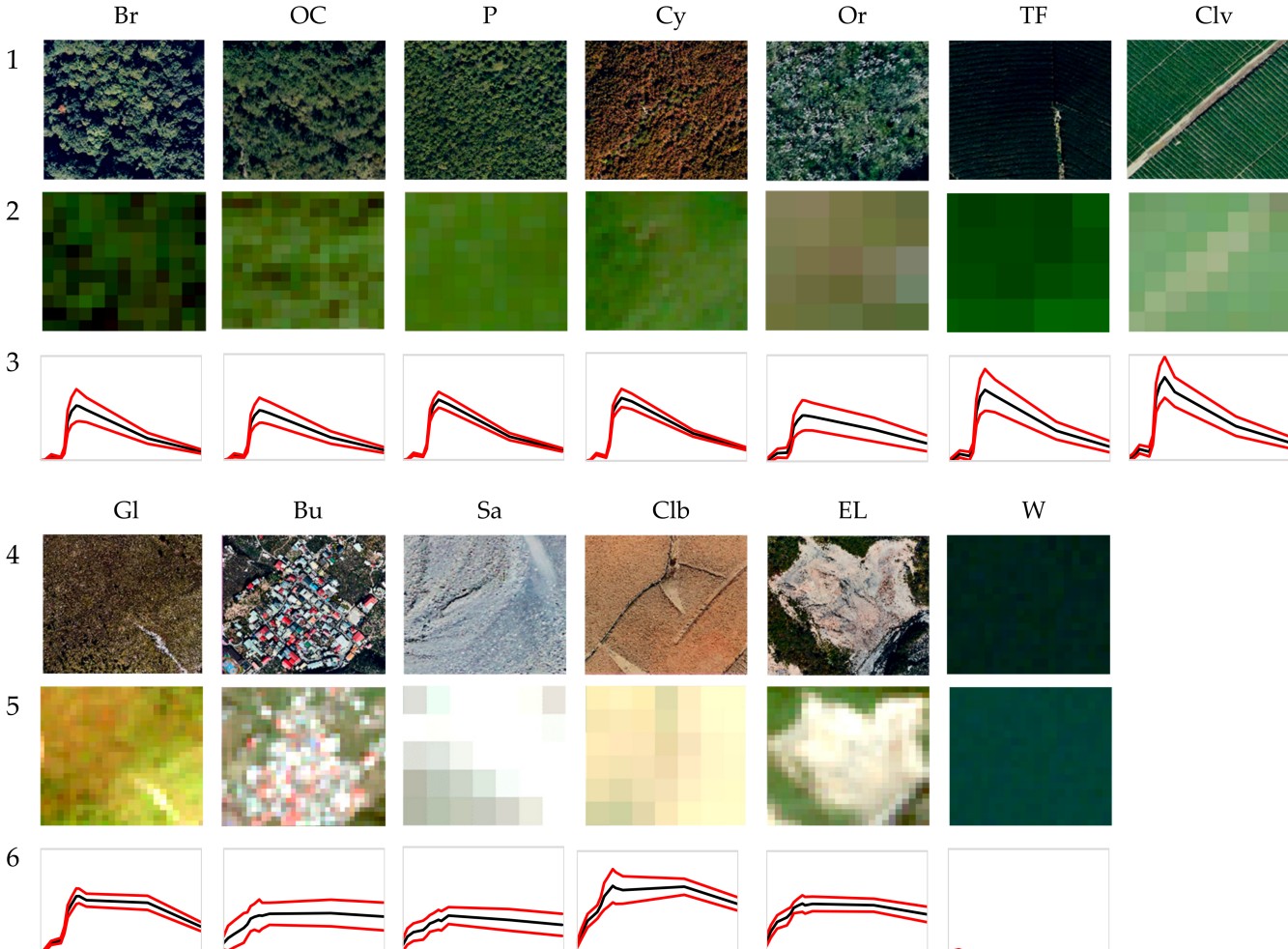

**Figure 5.** Variations of spectral feature for Level-III LULC classes. Images list in rows 1, 2, 3 and rows 4, 5, 6 represent high-resolution orthophoto image (1 and 4), Sentinel-2 image (2 and 5), and VNIR-SWIR spectral curve (3 and 6) (the cirrus band is excluded), respectively, of the specific class that came above them. Every spectral curve in rows 3 and 6 whose x-axis is with a range identical to the integrated line graph while the range of y-axis is extended to 0.52 to show the data variation. The black and red lines depict the reflectance curves in the form of $\mu \pm SD$, where $\mu$ and SD stand for the mean and standard deviation of reflectance, respectively. Please refer to Table 1 for the abbreviations of Level-III LULC classes.

### 3.2. Performance of Multi-Level LULC Classifications

Figure 7a–c show the Level-III, Level-II, and Level-I LULC thematic maps performed by the classifiers RF and SVM, respectively. Based on 80-20 training and test random partition process, the validation samples highlighted in Figure 1 were used to evaluate the classification performance. The details of the error matrix, including false-negative, false-positive, precision, recall, and F1-score, for the 13 end-members in Level-III classification of the study site simultaneously for RF and SVM are summarized in Table 2 and Figure 8. The precision and recall ranged from 47% to 100% meanwhile the F1-score was between 0.5 and 1.0 for all classes at the Level-III. The OA, kappa, and macro F1-score of the classifiers for the Level-III, -II, and -I are summarized in Figure 7d. It could be observed that the post-classification by attribute-based aggregation improved the accuracy from Level-III to Level-II, and eventually to Level-I for both the RF and SVM. The increase in

accuracy in Levels-II and -I is obviously contributed via the decrease in false-negative, false-positive in those classes that belongs to each of the forest, agriculture, and bare land categories. This indicates that both RF and SVM can address the attributes of LULC via multispectral features, their vegetation index derivatives, canopy height, and terrain slope. The comparison of the performance of the two classifiers revealed that the SVM achieved better OA, kappa, and macro-F1 score than the RF (Figure 7d). The area coverage of the thematic maps provided by the classifiers are significantly different for both the vegetated and non-vegetated classes (Table 3).

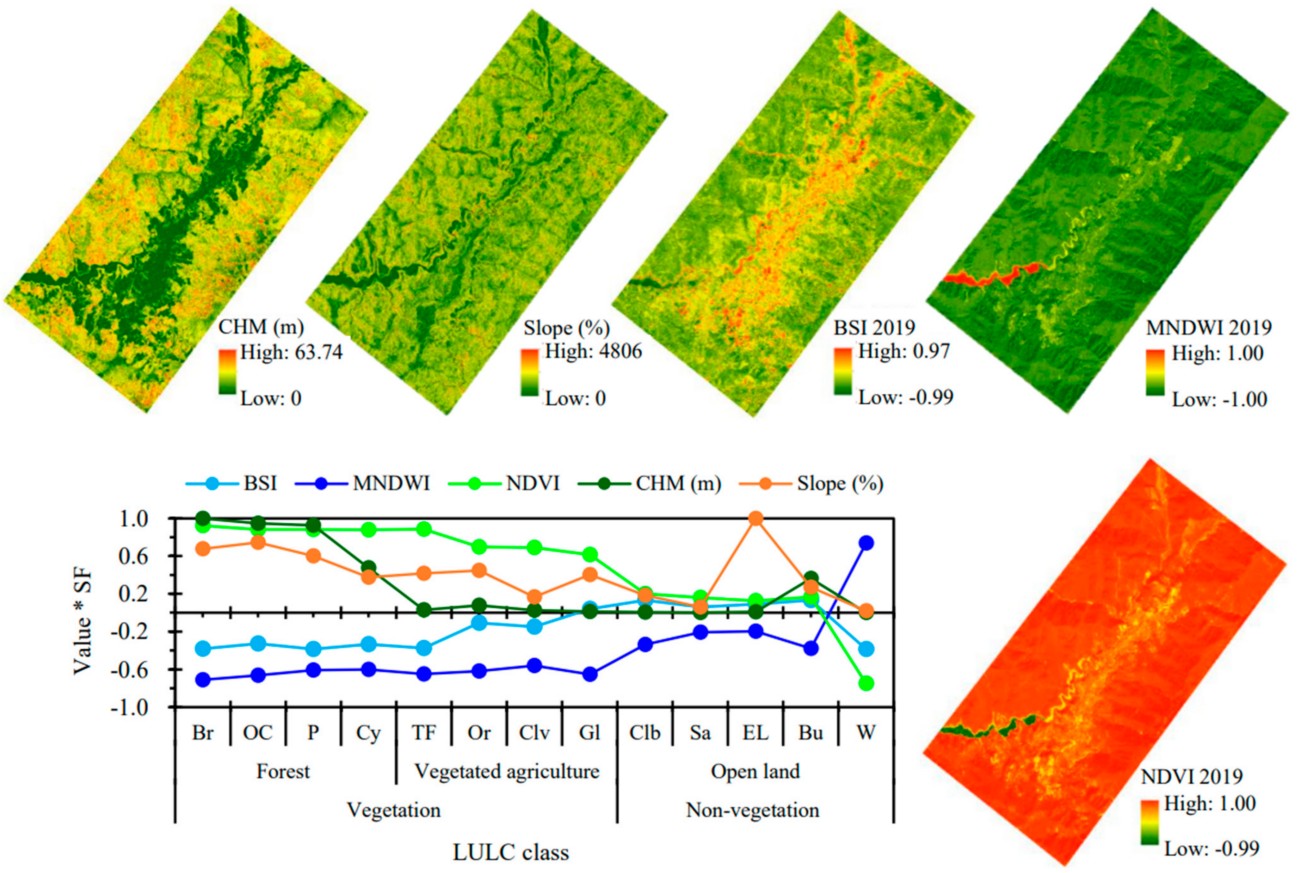

**Figure 6.** Sentinel-2 and ALS metric maps and average values of the Level-III LULC classes. The scale factor (SF) in the y-axis is 22.41 for CHM, 107.35 for slope, and 1 for BSI, MNDWI, and NDVI. The metrics revealed a dissimilarity among the classes beneficial for generating the RF model. Refer to Table 1 for the meaning of class code in the line graph.

**Table 2.** A joined confusion table of RF and SVM classifiers * at the Level-III LULC classification.

| | Class | Br (356) | | Cy (33) | | P (1366) | | OC (396) | | Gl (31) | | Or (744) | | TF (215) | | Clv (846) | | Clb (375) | | Bu (20) | | EL (38) | | Sa (140) | | W (794) | |
|---|---|---|---|---|---|---|---|---|---|---|---|---|---|---|---|---|---|---|---|---|---|---|---|---|---|---|---|
| | Br | 0.73 | 0.85 | 0.09 | 0.03 | 0.04 | 0.03 | 0.03 | 0.06 | | | 0.01 | 0.04 | | | 0.09 | | 0.01 | 0.20 | | | | | | | | |
| | Cy | | | 0.52 | 0.73 | 0.01 | | | | | | 0.01 | | | | 0.01 | | | | | | | | | | | |
| | P | | 0.09 | 0.09 | 0.12 | 0.86 | 0.88 | 0.26 | 0.19 | | | 0.01 | | 0.04 | 0.04 | 0.01 | 0.02 | | | | | | | | | | |
| | OC | | 0.06 | 0.18 | 0.12 | 0.07 | 0.05 | 0.68 | 0.74 | | | 0.003 | | | | 0.05 | 0.02 | 0.02 | | | | | | | | | |
| Predicted | Gl | | | | | | | | | 0.65 | 0.84 | | | 0.01 | | 0.01 | | 0.01 | 0.20 | | | | 0.10 | | | | |
| | Or | 0.22 | | 0.12 | | 0.01 | 0.01 | 0.02 | | 0.19 | 0.16 | 0.88 | 0.78 | 0.36 | 0.07 | 0.07 | 0.23 | 0.02 | 0.03 | 0.05 | | | | | | 0.07 | |
| | TF | 0.03 | | | | 0.01 | 0.03 | | | | | 0.004 | 0.04 | 0.55 | 0.73 | 0.002 | | 0.01 | | | | | | | | | |
| | Clv | 0.02 | | | | 0.01 | | | | 0.16 | | 0.04 | 0.11 | | | 0.66 | 0.75 | 0.02 | 0.08 | | | | | | | | |
| | Clb | | | | | | | | | | | 0.004 | 0.03 | | | 0.04 | | 0.72 | 0.77 | 0.05 | | | 0.11 | 0.10 | 0.04 | | |
| | Bu | | | | | | | | | | | | | | | | | 0.03 | | 0.70 | 0.85 | | | | | | |
| | EL | | | | | | | | | | | | | | | | | 0.01 | | 0.10 | 0.05 | 0.47 | 0.79 | 0.11 | 0.20 | | |
| | Sa | | | | | | | | | | | | | | | | | | | | | 0.53 | | 0.64 | 0.89 | | |
| | W | | | | | | | 0.01 | | | | 0.04 | | 0.005 | | 0.22 | | 0.21 | | | | | | 0.26 | | 1.0 | 1.0 |

* Yellow and green columns present the RF and SVM classifiers, respectively. The number in parenthesis comes after LULC class name indicating the number of validation pixels for the corresponding class.

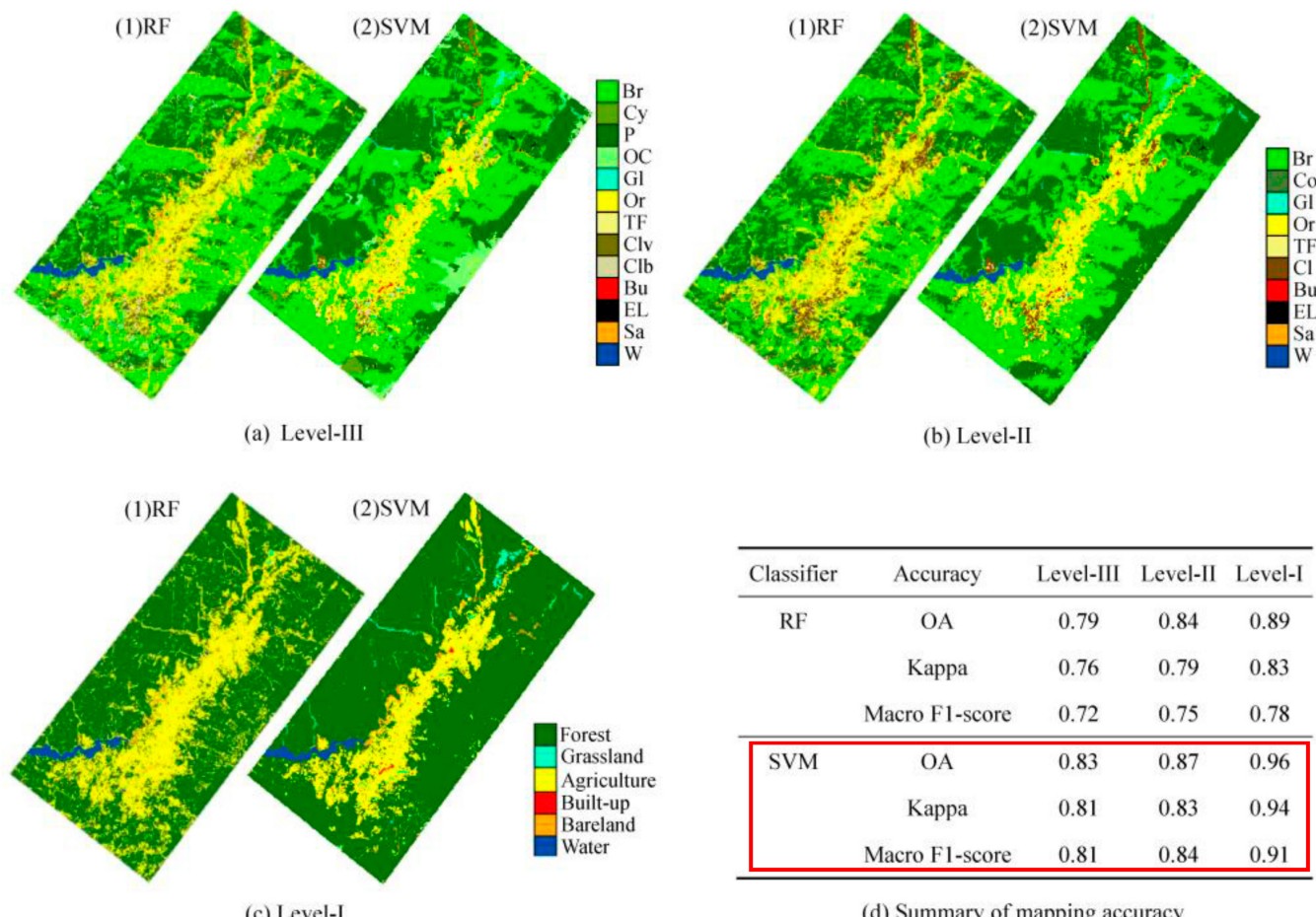

**Figure 7.** LULC classified images of the study site derived by RF and SVM approaches: (**a**) Level-III, (**b**) Level-II, (**c**) Level-I, and (**d**) summary of the classification accuracy. Please refer to Table 1 for the abbreviations of classes' legends for Level-III LULC. For the legend of Level-II LULC, the codes Co stands for conifer which is aggregated from Cy, P, and OC in Level-III; Cl indicates cropland including Clv and Clb in Level-III.

**Table 3.** Comparison of the area coverage (%) of each thematic class in each level provided by the classifiers.

| Level-III | | | Level-II | | | Level-I | | |
|---|---|---|---|---|---|---|---|---|
| **Class** | **RF** | **SVM** | **Class** | **RF** | **SVM** | **Class** | **RF** | **SVM** |
| Br | 36.37 | 44.64 | Br | 36.37 | 44.64 | F | 66.53 | 79.02 |
| Cy | 0.02 | 0.77 | Co | 30.15 | 34.39 | | | |
| P | 26.89 | 29.46 | | | | | | |
| OC | 3.24 | 4.16 | | | | | | |
| Gl | 0.07 | 1.54 | Gl | 0.07 | 1.54 | Gl | 0.07 | 1.54 |
| Or | 23.80 | 13.91 | Or | 23.80 | 13.91 | Ag | 31.14 | 16.95 |
| TF | 2.38 | 0.76 | TF | 2.38 | 0.76 | | | |
| Clv | 4.40 | 1.73 | Cl | 4.96 | 2.28 | | | |
| Clb | 0.55 | 0.55 | | | | | | |
| Bu | 0.004 | 0.19 | Bu | 0.004 | 0.19 | Bu | 0.004 | 0.19 |
| EL | 0.001 | 0.22 | EL | 0.001 | 0.22 | Bl | 0.94 | 1.16 |
| Sa | 0.94 | 0.95 | Sa | 0.94 | 0.95 | | | |
| W | 1.33 | 1.14 | W | 1.33 | 1.14 | W | 1.33 | 1.14 |

The colored box highlights the subclasses aggregated to a lower level of LULC class.

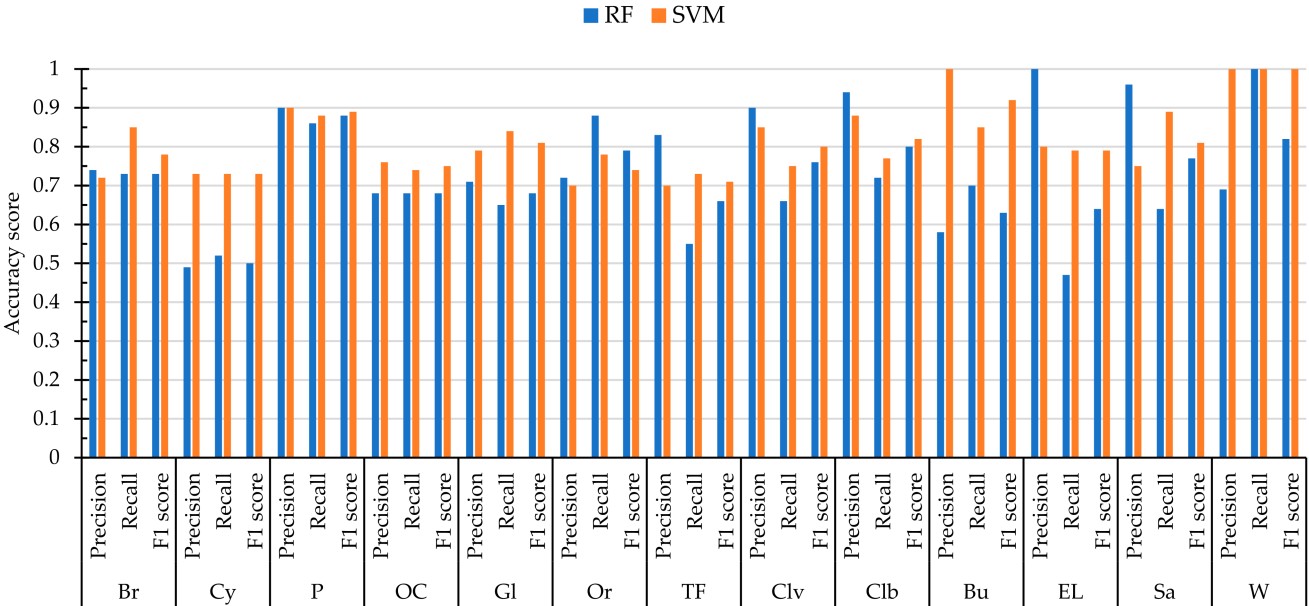

**Figure 8.** A paired-bar chart for comparison of precision, recall, and F1-score in every class at Level-III LULC classification obtained by RF and SVM. As noted, the two classifiers appeared equal efficiency in pine, while SVM achieved reliable classification in cypress (Cy), other conifer (OC), grassland (Gl), and built-up (Bu) as it obtained all the accuracy measures substantially greater than RF. The strength of RF was its ability torecognize orchard more efficiently than SVM. In general, the capability of RF in classifying LULC is generally poorer than SVM based on F1-score of all classes.

## 4. Discussion

### 4.1. Inherited Complexity of Biophysical Properties May Induce Reflectance Variation of Endmembers

Land cover, water, soil, and vegetation can be considered the simplest raw materials. They can be divided into various subcategories based on their physical, chemical, usage, and even biological properties, thus formulating diverse land cover land use. LULC mapping is a key to investigating information regarding end-members' properties, distributions, and effects on the land to support appropriate management for sustainability. However, due to the collective effects of environmental and temporal factors, the end-members' spectral features tend to be more divergent. This is evident, particularly in the progress of physiological activities. The complexity is naturally inherited, resulting in precise LULC classification being more difficult.

As can be seen in Table 2 (Section 3.2) and Figure 8, the classes' precision and recall achieved by the RF method were significantly lower than SVM method. Specifically, the cypress and building have a precision of less than 60%, at the same time the classes with a recall of less than 60% were the cypress, tea farm, and eroded land. More evidently, the RF method appeared to fail in differentiating vegetation classes because a pronounced false-negative and/or false-positive percentage (>20%) occurred in the broadleaf, cypress, other conifers, grassland, tea farm, and vegetated cropland. Meanwhile, the significant misclassification also indicates the RF failed to distinguish the open land classes, such as bare cropland, building, eroded land, and sand. In particular, there was 22%, 21%, 53%, and 26% of vegetated cropland, bare cropland, eroded land, and sand that were misclassified as water which does not merely show a significant omission in the former four classes but also an evident commission error in the latter class. Looking at the high false-positive and false-negative errors of eroded land and sand classes in the RF approach, the better classification accuracy of the SVM approach in differentiating the two classes indicates that the RF unfortunately is not optimal. Figure 9 shows some examples of LULC classification discrepancy in classifiers and ground truth. For diagnosing the development of desertification and the transformation of global drylands [61], the RF remains challenging.

Images in rows (1–3) show the orthoimage and RF and SVM classified Level-III images for (a) eroded land, (b) grassland, (c) built-up, and (d) sand; (e) the legend of LULC classes.

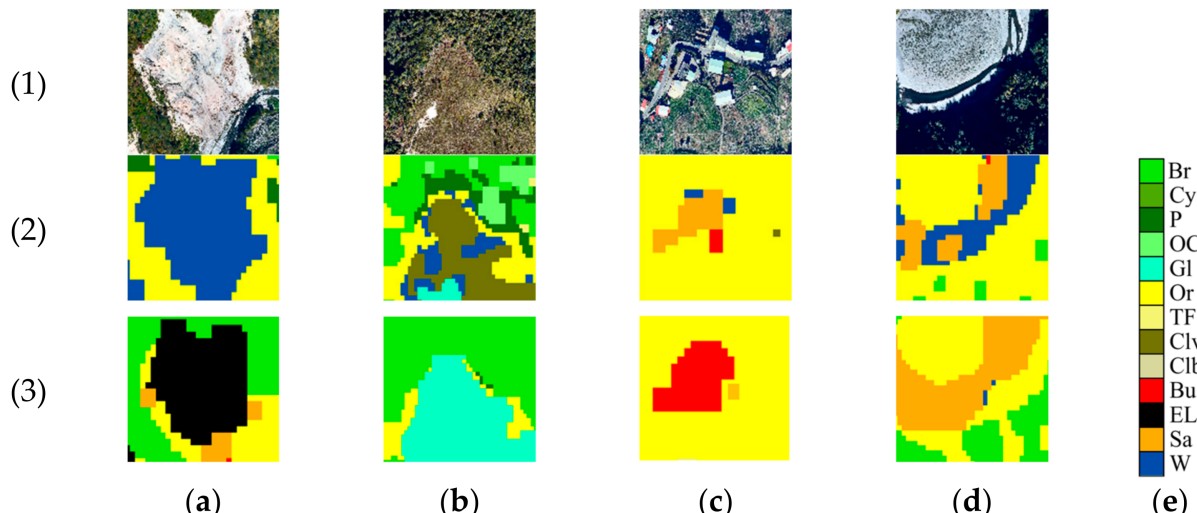

**Figure 9.** A close look at examples of LULC classification discrepancy in classifiers and ground truth. Images in rows (1–3) show the orthoimage and RF and SVM classified Level-III images. In (**a**), the eroded land was misclassified as water by RF but eroded land as it is by SVM. (**b**) is composed of grassland and other conifer which is classified as water, vegetated cropland, pine, broadleaf, and orchard by RF but mostly accurately classified as they are by SVM. (**c**) has built-up on the center and surrounded by orchard. Both RF and SVM recognized the orchard successfully; the built-up was mostly recognized as it is by SVM but mostly misclassified as sand by RF. (**d**) the sand was not correctly recognized by RF and SVM. (**e**) the legend of LULC classes.

### 4.2. Challenges in Deriving Robust Random Forest

In general, RF technique generates uncorrelated decision trees by random sampling with replacement from sampling dataset. Numerous independent decision trees that integrate features with appropriate decision nodes and homogeneous leaf nodes are formulated based on the bagging (or bootstrap aggregating) and splitting procedures. It is known that noise in training dataset may decrease the capability of decision trees in classification. Fortunately, the noise effect can be minimized by a feature importance tuning process, in which the least important features that contribute insignificant marginal classification accuracy will be removed from the decision tree generation. In addition, RF is flexible in generating deep models through increasing the number and depth of decision trees. However, the former increases voting accuracy with significant time cost while the latter tends to result in overfitting and reducing the accuracy or aggregation efficiency.

Based on the work demonstrated in this study, an interesting challenge might be to apply the RF technique to LULC classification as follow:

1. Considerable reflectance variations in vegetation classes increase the RF model's prediction uncertainty.

Supervised classification involves a process of image interpretation, objects or end-members of interest determination and sampling, spectral feature learning and modeling, and finally labeling and classification. For LULC mapping, the process can be flexible in view of the details of objects to be extracted from images for land and resources management. The Sentinel-2 spectral signatures of broadleaf, conifer, and agricultural classes observed in this study appeared to overlap each other partially. A similar condition of spectral signatures overlap also occurred in built-up and bare soil relevant classes. Obviously, this kind of spectral confusion has raised an issue of model reliability or prediction uncertainty of the RF model because node splitting for generating decision trees with a lowest impurity index becomes troublesome. Increasing complexity of LULC categories [62] and

vegetation phenology [63,64] will also increase the challenges in applying RF in achieving accurate LULC mapping.

2. Complexity and difficulty of LULC classification increase as the number and homogeneity of classes to be dealt with increases.

As noted, the 13 classes of interest at Level-III LULC classification have similar spectral signatures, and most are characterized as vegetation and bare soil materials. The homogeneity in the spectral dataset is moderately mitigated by incorporating canopy height, slope, and spectral indices BSI, NDVI, and MNDWI. The RF model with 500 decision trees was generated for classification. As can be seen in Figure 10, the error rate gradually reduced and reached a stable level as trees increased meanwhile the OOB error rate reached the lowest level when 16 features and 500 decision trees were assigned for the machine learning. The vegetation-relevant classes appeared with higher error rates in the modeling training and lower precision and recall accuracy in the validation dataset as well. For the best learning result, the RF model can characterize the Level-III LULC map with a macro F1-score of 0.72, increasing to 0.75 for the 10 classes Level-II map and 0.78 for the 6 classes Level-I map. This demonstrates that aggregating the higher-level LULC classes to generate a lower-level LULC map is feasible.

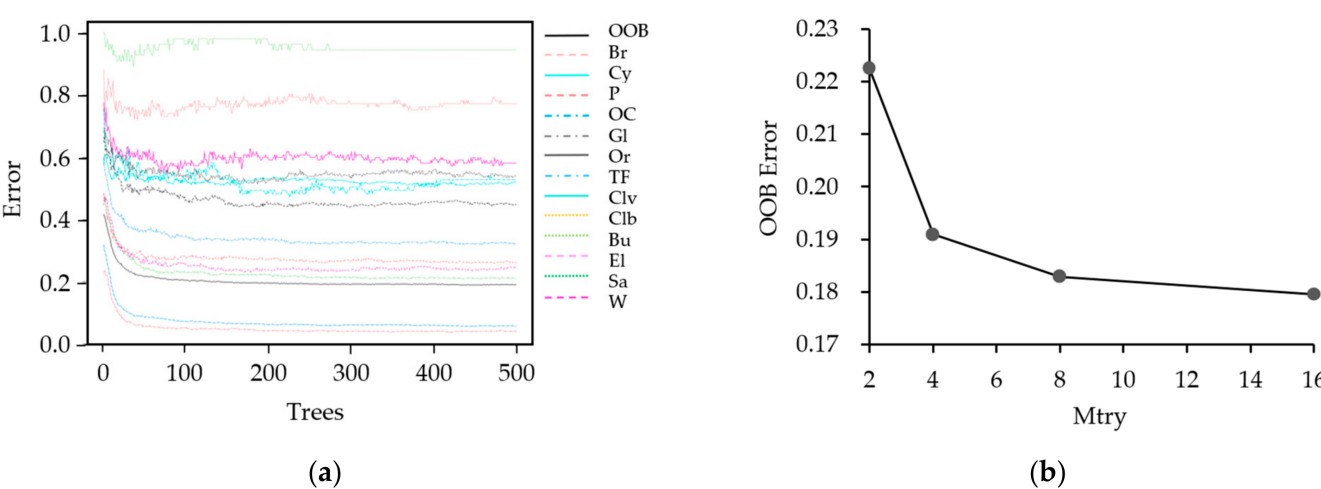

(**a**)          (**b**)

**Figure 10.** The trend of error rate changes in the process of generating entire RF model: (**a**) class-based and OOB error rate before tuning; and (**b**) the OOB error rate reduction achieved by tuning process.

3. High-level LULC classification with complicated and homogeneous classes seems to require a flexible non-linear model to derive reliable information.

In contrast to the performance of the RF approach, the SVM approach can formulate a more appropriate hyperplane with the same training dataset to achieve better accuracy based on the same evaluation dataset. The macro F1-score obtained was 0.81, 0.84, and 0.91 for Level-III, -II, and -I LULC, respectively, the improvement for each of the 3-level maps was 0.09, 0.09, and 0.13. Since the SVM modeling was implemented with the radial basis function, the excellent performance of SVM indicates that a non-linear-based hyperplane should be the key to achieving acceptable LULC classification. Recall that the generation process of an RF model mainly relies on node splitting and the structure of multiple decision trees, a method that can control the complicated non-linear feature space of LULC classes is expected.

4. Looking for possible ways to improve RF modeling for LULC classification.

RF technique can deal with quantitative prediction and qualitative classification. For the prediction, the response variable is numerical, and the outcome is determined as the average of all estimates from each of the multiple decision trees. However, the classification is a probability-based voting system for non-numerical class labeling, and the outcome

is determined as the majority of predictions. The learning process of developing the entire structure of a random forest is considered RF modeling. In general, a classification dealing with multiple classes requires more decision trees [65]. In an RF model without a sufficient number of decision trees and informational features, the accuracy measures such as precision, recall, and F1-score of interested classes will definitely not be acceptable, and therefore macro F1-score is also not satisfied.

Efficient management of land and natural resources needs accurate and reliable LULC information. This is particularly sensitive and important in the ecotone area, which is generally a mosaic of both rural and forest landscapes and whose land cover land use may frequently change due to anthropogenic activities and disturbances. Accurate mapping of LULC helps monitor the changes and moreover diagnose the drivers of corresponding changes. It is critical to constantly collect reliable LULC information as a part of FAO forest resources assessment for sustainable management.

An effective RF modeling must be able to differentiate LULC classes with nearly the same features as vegetative LULCs, with a large amount of decision trees expected as one potential key to accurate LULC mapping. Again, as observed in Figure 10a, the RF attained the lowest error when the trees were increased to 500. In addition, the integration of multispectral image, derived spectral indices, and ALS-based metrics is a potential way to improve RF modeling. To increase data dimensionality, multi-temporal images particularly can extend the spectral features at a single image to multiple images from a particular date to time serial dates. Consequently, the augmented dataset provides more opportunity to generate multiple decision trees with heterogeneous spectral features, which helps increase the number of decision trees while decreasing the risk of accuracy reduction or over-fitting problem. Accordingly, the generated multiple decision trees model is likely more robust, and the RF model is expected to reduce diversity of prediction outcome while raising the confidence of voting.

## 5. Conclusions

LULC classification is critical work for land and forest resources management. This is particularly significant for regular monitoring and assessment programs conducted by FAO. For example, the change of forest area, which can be monitored through multi-temporal LULC monitoring, is an indicator of sustainability. In addition, forest degradation and deforestation can be detected by extracting detailed attributes of forest types and species distribution which can also be provided by time-series LULC classification. Improving LULC classification helps land/resources management in an efficient way. The random forest technique has been well-developed and widely applied to LULC classification globally and has been reported for its excellent performance to this end. Nevertheless, challenges are still present when using the classifier in LULC mapping. The major challenge being faced by the RF which were revealed in this study is accurately differentiating different LULC classes with similar features. Although increasing the data dimensionality by integrating ancillary data can reduce the problem, it is still insufficient to completely improve the performance of the classifier. Given the findings of this study, additional techniques or modifications are still needed to fill the knowledge gap of accurate and high thematic resolution LULC mapping using the RF classifier.

**Author Contributions:** Conceptualization and methodology, C.L. and N.D.D.; resources, C.L.; formal analysis, N.D.D.; investigation, visualization, and writing—draft preparation, N.D.D. and C.L.; writing—review and editing, N.D.D. and C.L. All authors have read and agreed to the published version of the manuscript.

**Funding:** The Ministry of Science and Technology, Taiwan, funded this research—grant number MOST 108-2119-M-415-002 and MOST 109-2221-E-415-003. Airborne lidar data was acquired by the project funded by Forestry Bureau, Taiwan—grant number TFBC-1060113.

**Data Availability Statement:** The data presented in this study are available on request from the corresponding author. Due to data licensing, the lidar data are not publicly available.

**Acknowledgments:** The authors would like to thank the Lishan Office of Forestry Bureau staff for their assistance in the ground survey.

**Conflicts of Interest:** The authors declare no conflict of interest.

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
