# Peer review of "Challenges of Retrieving LULC Information in Rural-Forest Mosaic Landscapes Using Random Forest Technique"

_forests, doi:10.3390/f14040816_

Round 1

Reviewer 1 Report

The manuscript by Lin and Doyog presents a LULC mapping study for a landscape in Taiwan. The study compares random forest (RF) and support vector machine (SVM) methods as approaches for mapping LULC at three organizational levels (Level-I at the coarsest thematic resolution, Level-III at the finest). The analysis, including data preparation, modeling, and validation, follow common methods used in the literature. The study found that SVM outperformed RF and that accuracy generally increased from Level-III to Level-I LULC classifications, most likely because of challenges in differentiating spectrally similar LULC classes, which become more frequent as thematic resolution increases. The authors then relate their results to general challenges in LULC mapping and classification. The methods and results support many of the discussion points, though some statements could be better support (see comments below). The logic of the manuscript generally flows well, allowing the reader to follow the authors’ points.

However, I did find some key deficiencies in the manuscript, mostly related to the scope and framing of results, discussion, and conclusions.

(1)    Manuscript scope. This study deals with a case study comparison on two approaches for generating LULC classification maps in Taiwan. However, the title makes it sound like the paper deals more broadly with the challenges of random forest modeling. I think that disconnect is misleading. The abstract does not provide the details of the region being examined or the breadth of LULC classes being assessed. The last sentence of the abstract does not provide a key take home message for the study, just the identification of a single class where RF performed better that SVM (perhaps the previous sentence is more the conclusion statement?). Furthermore, the authors focus on LULC change in the conclusions, but they do not directly assess change in their landscape or accuracy of change mapping (which is different from accuracy of mapping LULC itself). I suggest that the authors carefully consider the scope of their study, the inferences produced, and the key messages, then rewrite the title and abstract to reflect that properly.  

(2)    Knowledge gap. The authors need to better describe the knowledge gap that they are filling. It appears that they are pointing out that a lot of RF modeling has been done (Lines 77-82), but perhaps not with a thematic resolution needed for forest monitoring in a rural landscape (Lines 98-103). If that is the case, the authors probably need to provide additional literature to support that statement. I know of papers that have LULC classes similar to what these authors present (e.g., Saah et al. 2020; https://www.sciencedirect.com/science/article/pii/S0303243419306270). As it stands, I think that many readers might not be convinced of the relevance of the contribution of the paper.

(3)    Proper references in the discussion. Section 4.2 (Challenges in deriving robust random forest) has relatively few citations. That is in spite of the fact that many statements made are broadly reported elsewhere. Without at least some citations used as examples of specific concepts (e.g., the ability of multispectral data to differentiate between LULC classes), the text reads as conclusions having arisen from their research alone. However, several components of the current study are largely confirmatory and need to be compared with other literature as appropriate. Those citations do not need to be exhaustive, but they need to be present to place the work in the appropriate context. I am confident that the authors know of many such studies. Anyway, it seems to me that citations should be added and text revised for lines 360-432 to provide appropriate context.

Even with these major deficiencies, this paper can add to the literature highlighting factors that influence LULC classification accuracy, and thus map product applicability.

**Specific Comments**

Line 13: Sentinel-1 is radar, Senitnel-2 is multispectral data. Methods indicate that Sentinel-2 was used. I also see this issue in line 366.

Line 14: This implies that Sentinel-2 provides slope and canopy height. It looks like the height and slope information were generated from lidar, so that should be stated. In fact, the authors could note the integration of multispectral imagery and lidar as a strength

Line 20: “complex” and “homogeneous” are nearly antonyms, so I found this confusing

Line 21: Replace “changeable” with “challenging”?

Line 23: There appears to be something missing after “built-up” as the sentence ends abruptly

Lines 35-39. These two sentences both start with “It”, but I am not sure I know what the authors are referring to.

Line 40: What do the authors mean by “interferences”. Is it just being used to mean change. If that is the case, it seems repetitive to state that LULC change indicates ecosystem change. Consider rewording this sentence to clarify.

Lines 75-77: I do not know what this sentence is trying to say

Lines 120-121: Seems like there is some direction confusion here. The National Park is referred to as being to the left (west), but then Taichung is referred to as being west. I assume one of these is east? Also, I would stick with cardinal directions (east, west, north, and south) as opposed to left, right, up, or down.

Line 124: Agriculture is always influenced by agricultural factors, so I assume this is referring to the fires. Can the authors make this clearer?

Lines 124-125: Is a fire only 2.5 hectares in size really worth mentioning? That is just 62 Sentinel-2 pixels. Maybe the units are wrong?

Line 135: Replace “atmospheric” with “atmospherically”

Line 131: The authors have two nice workflow figures (Figures 2-3). Figure 2 is not referenced in the text and should probably be referenced at the beginning of this section. Perhaps a sentence to start this section that references Figure 2 as the general workflow for data processing?

Lines 160-162: I suggest including point density, ground point density, scan angle, and scan overlap to provide information on the information quantity and quality in the lidar acquisition.

Lines 173-177: I suggest that the authors present a table that places the Level-I, Level-II, and Level-III in their hierarchy, so that the reader has for reference. Otherwise, it is not clear how the two relate. They are referred to later (Figure 7). This table could also include the abbreviations for each class.

Line 183: Kappa has been criticized (https://www.sciencedirect.com/science/article/pii/S0034425719306509). Perhaps choose a different measure of model skill, such as true skill statistics or at least explain why you consider it appropriate to use kappa here.

Line 233: Should this reference Figure 6, not Figure 5?

Lines 237-238: Not sure why the term “baseline” is introduced relative to the CHM. The authors could just stick with “zero” as the term they use as that is basically what the rescaled variable is indicating, correct?

Figures 5-6: I like both of these figures.

Line 260: There is no table 3 in the manuscript.

Figure 7d. I like this inset, but aren’t all of these metrics reported in the main text? I suggest that the authors could reduce the redundancy, either by removing 7d or by editing the text to focus on the main points (increasing accuracy and skill as thematic resolution decreases and as we shift from RF to SVM.

Line 291: This section title seems like speculation and is not directly addressed by the results. I suggest that the authors qualify this statement somehow, perhaps replacing “induced” with “may induce”. Alternatively, if the authors believe that their results explicitly examine complex biophysical properties, cite those figures and tables here.

Lines 301-306. This seems like results (as do Table 1 and Figure 8), so move this to methods. Figure 8 is repetitive of Table 1, so either delete Figure 8 or remove precision, recall, and F1-score from the table

Line 318: Replace “El” with “eroded land”

Line 323: I do not know what “changeable” means in this context

Lines 360-371: The reflectance variation contribution to uncertainty, portrayed here as overlap in spectral signals between different classes, really just boils down to a lack of information, correct? It is unclear to me why this issue is unique to RF models. Also, this issue has been discussed elsewhere (e.g., https://www.mdpi.com/2073-445X/10/9/994, https://www.mdpi.com/2072-4292/14/9/1977), so a few citations linking this challenge to the literature would be good.

Lines 421-423: This sentence seems to be missing a few words. Also, again the authors refer to complex and homogeneous (“less heterogeneous” in this case) vegetation, which seem to contrast (complex tends to be used to describe varied conditions, which would not be less heterogeneous, in general).

Lines 421-432: I found this paragraph hard to follow. Also, several of the points made are not addressed by results of this study, but are well represented in the literature. Here, and probably throughout the discussion, I found references to the literature to be generally lacking.

Line 434: Delete “always” as it does not really add anything here.

Lines 436-438: As far as I can tell, change LULC change was not assessed in the current study. It might help here to explicitly state that change can be derived from LULC mapping in multiple years. I know this is probably obvious, but making it clear that the authors do not actually address this in their paper is important.

Lines 441-452: This is mostly just a restatement of the study results with a few notes on implications. The conclusions need to focus on the implications of the study and the big “so what?”

Author Response

Dear Reviewer:

Thank you very much for your valuable time and comments. We have carefully checked each of the comments and made changes to the manuscript accordingly. The point-to-point response to each of your comments is attached.

Reviewer 2 Report

Dear Authors,

The paper entitled “Challenges of retrieving LULC information in rural-forest mosaic landscapes using random forest technique” integrated Sentinel-1 multispectral surface reflectance, vegetation indices, canopy height, and slope to generate a random forest model for 3-level LULC classification. This study aims to determine the supplementary abilities of the well-performing RF technique in establish forest composition-oriented Land Use Land Cover information. Material and method described very well. Results and the figures of this section are acceptable and complete. The discussion must revise very carefully. In this section the authors did not discuss their results and did not compare with others. On the other hand, the conclusion supports the results. The paper must consider after major revision. Also some of the general comments as follow:

1.       Abstract: 3-level LULC classification or Level-III classification? Which one?

2.       Introduction: In this section, it is necessary to use valuable information of some papers specially in the first and second paragraphs. Use The following paper:

https://doi.org/10.3390/land11010006

3.       LN 105: this study's objective 105 is toàthis study aims to….

4.       Fig.1: Use a and b instead of left and right for the figure and explain in the caption. Also, the quality is poor and replace it with a better one.

Author Response

(The authors gave the same response as above.)

Reviewer 3 Report

The paper presents the results of a study aimed at classifying land use and land cover using Sentinel 2 and LiDAR data and machine learning techniques (RF and SVM).

The work does not present substantial methodological novelties; the lack of details on the tested classification system (just named LULC Level III, II and I) and on the characteristics of the study area (the extension and characteristics of the different classes are missing) does not allow to evaluate the results.

The methods section is very incomplete and confusing. There is a complete lack of information on the ground truth data, on how they were selected and classified and then divided between training and validation (apart from a mention in the discussion paragraph). Details on LiDAR data are also missing, Sentinel data is called Sentinel 1 data twice instead of Sentinel 2 (lines 13 and 366) and it is non clarified why the authors used Sentinel-2B products and not the available BOA products L2A. RF is described in detail but a discussion of SVM is totally missing. A description in formula or a bibliographic reference on the accuracy indices used would have been very useful to help understanding the text.

The article is not properly structured, some important information about the methods are reported in the paragraphs dedicated to the results or the discussion (e.g. lines 229-236, 267-269, 380-81, 396-97), part of the results are reported in the discussion section (e.g. the error matrix of Table 1 and figs. 8 and 9).

There are also serious shortcomings of a methodological nature. For example, the accuracy of the classification for the higher hierarchical levels should be evaluated on the results obtained by applying RF and SVM directly on training data classified according to the level of interest (level I or II) and not as the authors do by aggregating afterwards the points attributed by RF/SVM to the level III classes. The confusion matrix provides percentage values equivalent to the producer’s accuracy (which however is not mentioned) rather than the number of points/pixels.

Other minor problems concern the correspondence between figures and text and the quality of the figures: fig. 2 is not mentioned in the text, in fig. 5 the units of measurement in the graphs are missing, the table 3 mentioned in the caption of figure 6 does not exist and a table with of the LULC classes abbreviations is missing.

In conclusion, I suggest re-submit the work after having improved its structure and consistently integrated the part describing the data and the applied methods. It is also necessary to apply the two machine learning techniques independently to the different hierarchical levels investigated. Finally, it is suggested to focus discussions on the actual results of the work and not on general concepts as in the current version.

Author Response

(The authors gave the same response as above.)

Reviewer 4 Report

Dear Authors:

   Your article: “Challenges of retrieving LULC information in rural-forest mosaic landscapes using random forest technique” presents a recurring theme that refers to the automated classification of LULC. After a thorough analysis of your article, I proposed minor recommendations that can be consulted in the comments on the digital file. Still, in the article, the use of a weighted Kappa index tends to better explain the results obtained. (Olofsson, P., Foody, G. M., Herold, M., Stehman, S. V., Woodcock, C. E., & Wulder, M. A. (2014). Good practices for estimating area and assessing the accuracy of land change. Remote Sensing of Environment, 148, 42–57. https://doi.org/10.1016/j.rse.2014.02.015). It would be interesting for you to present the area of each thematic class in each level. Another issue to be explored: why not use a layer resulting from the segmentation of Sentinel 2 images? This tends to decrease the salt and pepper effect seen in classified images. Another issue to be analyzed: with many variables, wouldn't it be interesting to implement the Recursive Feature Elimination technique to select the best predictor variables? This is an important step because an excess of predictor variables tends to make the RF less efficient. I would also like your attention to the following observations:

1) Lines 95 to 102: Review the wording. Extensive and confusing text.

2) Figure 1: Additional base map showing the cited boundaries between lines 119 and 121.

3) How big is the study area? How was the limit shown defined?

4) Line 136: “and then resampled or pan-sharpened”: which of the two options? Please mention the methods for resampling or pan-sharpened.

5) Please provide more characteristics of the point cloud, such as density and precisions associated with the sensor.

6) Present the criteria for defining the spatial resolution of 1m for the DEM and DSM. Which parameters are used to filter Lidar data?

7) Line 170: “As a result, a data cube with 17 features” Did you create a data cube or did you create a raster file with 17 layers (stacked data)?

8) Detail the hardware resources and computational applications used in the data processing. You can mention the applications in the processing flowchart.

9) Line 168: Detail the sensor characteristics. Would it be GEDI? In this case, you interpolated a surface. Detail the processing steps. Usually, Gedi data has a much larger spacing of footprints when compared to data derived from an ALS.

10) Line 179: “26769 sample pixels covering the 13 Level-III LULCs were randomly selected and separated into two datasets for modeling and validation”. Present the methodology used for sample selection. Quote here the ratio training and testing (80/20).

11) Figure 4: Delimit the area of Figure 4b in Figure 4a.

12) Line 233: “The Slope and CHM are rescaled to 0 – 1 by dividing a rescaling factor (SF) of 107.35 and 22.41, respectively”. Explain this normalization criterion.

13) Line 268: “aggregation according to attributes of classes as indicated”. Explain the methodology for aggregation, as well as the parameters used to configure the algorithm.

14) Line 278: Display the confusion matrix shown in table 1 in the results chapter.

15) Figure 7d: Highlight the best results obtained in bold.

16) Line 321: “discrepancy between the orthoimage (ground truth) and classification results of RF and SVM” Did you use the samples (20%) or the orthoimage for validation?

17) Confront the results obtained with those of the authors cited in the bibliographical references, thus subsidizing the discussion presented in 4.2.

18) Conclusions: results and citations were repeated. It would be interesting in this chapter to answer: were the objectives achieved? What are the recommendations for future studies?

   I end my review by congratulating them for the work and article presented.

Respectfully,

Author Response

(The authors gave the same response as above.)

Round 2

Reviewer 2 Report

Good job!

Author Response

Dear Reviewer,

Thank you very much for your recommendation and valuable time.

Chinsu

Reviewer 4 Report

Dear authors,

       The second version of your manuscript incorporated most of my suggestions, but this question and its answer:

       Question: Detail the hardware resources and computational applications used in the data processing. You can mention the applications in the processing flowchart.

       Response: With the advancement of technology, several platforms can be used in performing ML-based classification like R, ENVI, SNAP, Arcgis pro, and python. The researcher can choose which to use depending on his convenience so we did not specify the software that were used in this study.

       I requested this information to facilitate the replication of your methodology by other authors and thus facilitate the citation of the study elsewhere. I particularly use all the applications you mentioned, especially Python. The problem is that novice researchers may find it difficult to properly identify the tools used and, as a result, the learning and execution curve ends up being costly and sometimes demotivating.

         After completing the reading of the new version of the article, comparing it with the first one, it can be seen that the main adjustments were incorporated.

         I finish my analysis by congratulating them for the new version of the article.

Sincerely,

Author Response

Dear Reviewer,

Thank you very much for expanding on the previous comment. The analysis was done via the library randomForest in R [Liaw and Wiener, 2002] and the tool SVM in ENVI. As requested, the tool information is added to the flowchart in Figure 2.

Best regards,

Chinsu